# DELTA: DEep Learning Transfer using Feature Map with Attention for Convolutional Networks

**Xingjian Li[†], Haoyi Xiong[†], Hanchao Wang[†], Yuxuan Rao[†,‡], Liping Liu[+], Jun Huan[†]**
[†] Big Data Lab, Baidu Reaseach
[‡] University of Illinois at Urbana-Champaign
[+] Big Data Department, Baidu Inc.
`{lixingjian,xionghaoyi,wanghanchao,lipingliu,huanjun}@baidu.com`

## Abstract

Transfer learning through fine-tuning a pre-trained neural network with an extremely large dataset, such as ImageNet, can significantly accelerate training while the accuracy is frequently bottlenecked by the limited dataset size of the new target task. To solve the problem, some regularization methods, constraining the outer layer weights of the target network using the starting point as references (SPAR), have been studied. In this paper, we propose a novel regularized transfer learning framework DELTA, namely _DEep Learning Transfer using Feature Map with Attention_. Instead of constraining the weights of neural network, DELTA aims to preserve the outer layer outputs of the target network. Specifically, in addition to minimizing the empirical loss, DELTA aligns the outer layer outputs of two networks, through constraining a subset of feature maps that are precisely selected by attention that has been learned in a supervised learning manner. We evaluate DELTA with the state-of-the-art algorithms, including $L^2$ and $L^2$-$SP$. The experiment results show that our method outperforms these baselines with higher accuracy for new tasks.

## 1 Introduction

In many real-world applications, deep learning practitioners often have limited number of training instances. Direct training a deep neural network with a small training data set usually results in the so-called _over-fitting_ problem and the quality of the obtained model is low. A simple yet effective approach to obtain high-quality deep learning models is to perform weight fine-tuning. In such practices, a deep neural network is first trained using a large (and possibly irrelevant) source dataset (e.g. ImageNet). The weights of such a network are then fine-tuned using the data from the target application domain.

Fine-tuning is a specific approach to perform transfer learning in deep learning. The weights pre-trained by the source dataset with a sufficiently large number of instances usually provide a better initialization for the target task than random initializations. In a typical fine-tuning approach, weights in lower convolution layers are fixed and weights in upper layers are re-trained using data from the target domain. In this approach parameters of the target model may be driven far away from initial values, which also causes over-fitting in transfer learning scenarios.

Approaches called regularization using the starting point as the reference (SPAR), were recently proposed to solve the over-fitting problem. For example, Li _et al._ (Li et al., 2018) proposed $L^2$-$SP$ that incorporates the Euclid distance between the target weights and the starting point (i.e., weights of source network) as part of the loss. Minimizing this loss function, $L^2$-$SP$ aims to minimize the empirical loss of deep learning while reducing the distance of weights between source and target networks. They achieved significant improvement compared with standard practice of using the weight decay ($L^2$ normalization).

However such regularization method may not deliver optimal solution for transfer learning. On one side, if the regularization is not strong, even with fine-turning, the weights may still be driven far away from the initial position, leading to the lose of useful knowledge, i.e. catastrophic memory loss. On the other side, if the regularization is too strong, newly obtained model is constrained to a local neighborhood of the original model, which may be suboptimal to the target data set. Although aforementioned methods demonstrated the power of regularization in deep transfer learning, we argue that we need to perform research on at least the following two aspects in order to further improve current regularization methods.

*Behavior vs. Mechanisms.* The practice of weight regularization for CNN is motivated by a simple intuition — the network (layers) with similar weights should produce similar outputs. However, due to the complex structures of deep neural network with strong redundancies, regulating the model parameters directly seems an over-killing of the problem. We argue that we should regularize the "Behavior", or in our case, the outer layer outputs (e.g. the feature maps) produced by each layer, rather than model parameters. With constrained feature maps, the generalization capacity could be improved through aligning the behaviors of the outer layers of the target network to the source one, which has been pre-trained using an extremely large dataset. In Convolutional Neural Networks, which we focus on exclusively in this paper, an *outer layer* is a convolution layer and the *output* of an outer layer is its feature map.

*Syntax vs Semantics.* While regularizing the feature maps might improve the transfer of generalization capacity, it is still difficult to design such regularizers. It is challenging to measure the similarity/distance between the feature maps without understanding its semantics or representations. For example for image classification, some of the convolution kernels may be corresponding to features that are shared between the two learning tasks and hence should be preserved in transfer learning while others are specific to the source task and hence could be eliminated in transfer learning.

In this paper, we propose a novel regularization approach DELTA to address the two issues. Specifically, DELTA selects the discriminative features from outer layer outputs through re-weighting the feature maps with a novel supervised attention mechanism. Through paying attention to discriminative parts of feature maps, DELTA characterizes the distance between source/target networks using their outer layer outputs, and incorporates such distance as the regularization term of the loss function. With the back-propagation, such regularization finally affects the optimization for weights of deep neural network and awards the target network generalization capacity inherited from the source network.

In summary, our key insight is what we call "***unactivated channel re-usage***". Specifically our approach identifies those transferable channels and preserves such filters through regularization and identify those untransferable channels and reuse them, using an attention mechanism with feature map regularization.

We have conducted extensive experiments using a wide range of source/target datasets and compared DELTA to the existing deep transfer learning algorithms that are in pursuit of weight similarity. The experiment results show that DELTA significantly outperformed the state-of-the-art regularization algorithms including $L^2$ and $L^2\text{-}SP$ with higher accuracy on a wide group of image classification data sets.

The rest of the paper is organized as follows: in Section 2 related works are summarized, in Section 3 our feature map based regularization method is introduced, in Section 4 experimental results are presented and discussed, and finally in Section 5 the paper is concluded.

## 2 RELATED WORK AND BACKGROUNDS

In this section, we first review the related works to this paper, where we discuss the contributions made by this work beyond previous studies. Then, we present the backgrounds of our work.

### 2.1 RELATED WORK

Transfer learning is a type of machine learning paradigm aiming at transferring the knowledge obtained in a source task to a target task (Caruana, 1997; Pan et al., 2010). Our work primarily focuses on inductive transfer learning for deep neural networks, where the label space of the target task dif-

fers from that of the source task. For example, Donahue *et al* (Donahue et al., 2014) proposed to train a classifier based on feature extracted from a pre-trained CNN, where a large mount of parameters, such as filters, of the source network are reused directly in the target one. This method may overload the target network with tons of irrelevant features (without discrimination power) involved, while the key features of the target task might be ignored. To understand whether a feature can be transferred to the target network, Yosinki *et al.* (Yosinski et al., 2014) quantified the transferability of features from each layer considering the performance gain. Moreover, to understand the factors that may affect deep transfer learning performance, Huh *et al.* (Huh et al., 2016) empirically analyzed the features obtained by the ImageNet pre-trained source network to a wide range of computer vision tasks. Recently, more studies to improve the inductive transfer learning from a diverse set of angles have been proposed, such as filter subset selection (Ge & Yu, 2017; Cui et al., 2018), sparse transfer (Liu et al., 2017), filter distribution constraining (Aygun et al., 2017), and parameter transfer (Zhang et al., 2018).

For deep transfer learning problems, the most relevant work to our study is (Li et al., 2018), where authors investigated regularization schemes to accelerate deep transfer learning while preventing fine-tuning from over-fitting. Their work showed that a simple $L^2$-norm regularization on top of the "Starting Point as a Reference" optimization can significantly outperform a wide range of regularization-based deep transfer learning mechanisms, such as the standard $L^2$-norm regularization. Compared to above work, the key contributions made in this paper include 1) rather than regularizing the distance between the parameters of source network and target network, DELTA constrains the $L^2$-norm of the difference between their behaviors (i.e., the feature maps of outer layer outputs in the source/target networks); and 2) the regularization term used in DELTA incorporates a supervised attention mechanism, which re-weights regularizers according to their performance gain/loss.

In terms of methodologies, our work is also related to the knowledge distillation for model compression (Hinton et al., 2015; Romero et al., 2014). Generally, knowledge distillation focuses on teacher-student network training, where the teacher and student networks are usually based on the same task (Hinton et al., 2015). These work frequently intends to transfer the knowledge in the teacher network to the student one through aligning their outputs of some layers (Romero et al., 2014). The most close works to this paper are (Zagoruyko & Komodakis, 2016; Yim et al., 2017), where knowledge distillation technique has been studied to improve transfer learning. Compared to above work, our work, including other transfer learning studies, intends to transfer knowledge between different source/target tasks (i.e., source and target tasks), though the source/target networks can be viewed as teachers and students respectively. We follow the conceptual ideas of knowledge distillation to regularize the outer layer outputs of the network (i.e., feature maps), yet further extend such regularization to a supervised transfer learning mechanism by incorporating the labels of target task (which is different from the source task/network). Moreover, a supervised attention mechanism has been adopted to regularize the feature maps according to the importance of filters. Other works relevant to our methodology include: continual learning (Kirkpatrick et al., 2017; Li & Hoiem, 2017), attention mechanism for CNN models (Mnih et al., 2014; Xu et al., 2015; Yang et al., 2016; Zagoruyko & Komodakis, 2016), among others.

## 2.2 BACKGROUNDS

Deep convolutional networks usually consist of a great number of parameters that need fit to the dataset. For example, ResNet-110 has more than one million free parameters. The size of free parameters causes the risk of over-fitting. Regularization is the technique to reduce this risk by constraining the parameters within a limited space. The general regularization problem is usually formulated as follow.

### 2.2.1 GENERAL REGULARIZATION

Let's denote the dataset for the desired task as $\{(\mathbf{x}_1, y_1), (\mathbf{x}_2, y_2), (\mathbf{x}_3, y_3) \ldots, (\mathbf{x}_n, y_n)\}$, where totally $n$ tuples are offered and each tuple $(\mathbf{x}_i, y_i)$ refers to the input image and its label in the dataset. We further denote $\omega \in \mathbb{R}^d$ be the $d$-dimensional parameter vector containing all $d$ parameters of the

target model. The optimization object with regularization is to obtain

$$\min_w \sum_{i=1}^n L(z(\mathbf{x}_i, \omega), y_i) + \lambda \cdot \Omega(\omega) \qquad , \qquad (1)$$

where the first term $\sum_{i=1}^n L(z(\mathbf{x}_i, \omega), y_i)$ refers to the empirical loss of data fitting while the second term is a general form of regularization. The tuning parameter $\lambda > 0$ balances the trade-off between the empirical loss and the regularization loss. Without any explicit information (such as other datasets) given, one can easily use the $L^0/L^1/L^2$-norm of the parameter vector $\omega$ as the regularization to fix the consistency issue of the network.

### 2.2.2 REGULARIZATION FOR TRANSFER LEARNING

Given a pre-trained network with parameter $\omega^*$ based on an extremely large dataset as the source, one can estimate the parameter of target network through the transfer learning paradigms. Using the $\omega^*$ as the initialization to solve the problem in Eq 1 can accelerate the training of target network through knowledge transfer (Hinton et al., 2006; Bengio et al., 2007). However, the accuracy of the target network would be bottlenecked in such settings. To further improve the transfer learning, novel regularized transfer learning paradigms that constrain the divergence between target and source networks has been proposed, such that

$$\min_w \sum_{i=1}^n L(z(\mathbf{x}_i, \omega), y_i) + \lambda \cdot \Omega(\omega, \omega^*) \qquad (2)$$

where the regularization term $\Omega(\omega, \omega^*)$ characterizes the differences between the parameters of target and source network. As $\omega^*$ is frequently used as the initialization of $\omega$ during the optimization procedure, this method sometimes refers to Starting Point As the Reference (SPAR) method. To regularize weights straightforwardly, one can easily use the geometric distance between $\omega$ and $\omega^*$ as the regularization terms. For example, $L^2$-$SP$ algorithm constrains the Euclid distance of the weights of convolution filters between the source/target networks (Li et al., 2018).

In this way, we summarize the existing deep transfer learning approaches as the solution of the regularized learning problem listed in Eq 2, where the regularizer aims at constraining the divergence of parameters of the two networks while ignoring the behavior of the networks with the training data set $\{(\mathbf{x}_1, y_1), (\mathbf{x}_2, y_2), \ldots, (\mathbf{x}_n, y_n)\}$. More specific, the regularization terms used by the existing deep transfer learning approaches neither consider how the network with certain parameters would behave with the new data (images) or leverages the supervision information from the labeled data (images) to improve the transfer performance.

## 3 LEARNING FRAMEWORK AND ALGORITHMS

In this section, we first formulate the problem, then present the overall design of proposed solution and introduce several key algorithms.

### 3.1 OVERALL FRAMEWORK

In our research, instead of bounding the difference of weights, we intend to regulate the network behaviors and force some layers of the target network to behave similarly to the source ones. Specifically, we define the "behaviors" of a layer as its output, which are with semantics-rich and discriminative information.

DELTA intends to incorporate a new regularizer $\Omega'(\omega, \omega^*, \mathbf{x})$. Given a pre-trained parameter $\omega^*$ and any input image $\mathbf{x}$, the regularizer $\Omega'(\omega, \omega^*, \mathbf{x})$ measures the distance between the behaviors of target network with parameter $\omega$ and the source one based on $\omega^*$. With such regularizer, the transfer learning problem can be reduced to learning problem as follows:

$$\min_w \sum_{i=1}^n L(z(\mathbf{x}_i, \omega), y_i) + \sum_{i=1}^n \Omega(\omega, \omega^*, \mathbf{x}_i, y_i, z) \qquad (3)$$

where $\sum_{i=1}^n \Omega(\omega, \omega^*, \mathbf{x}_i, y_i, z)$ characterizes the aggregated difference between the source and target network over the whole training dataset using the model $z$. Note that, with the input tuples

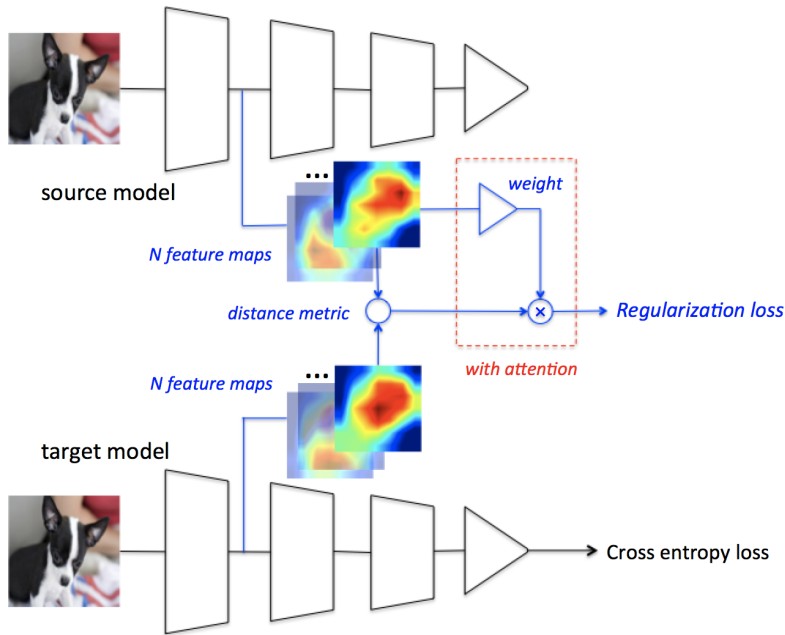

Figure 1: Behavior-based Regularization using Feature Maps with Attentions

$(\mathbf{x}_i, y_i)$ and for $1 \leq i \leq n$, the proposed regularizer $\Omega(\omega, \omega^*, \mathbf{x}_i, y_i, z)$ is capable of regularizing the behavioral differences of network model $z$ based on each labeled sample $(\mathbf{x}_i, y_i)$ in the dataset, using the parameters $\omega$ and $\omega^*$ respectively.

Further, inspired by SPAR method, DELTA accelerates the optimization procedure of the regularizer through incorporating a parameter-based proximal term, such that

$$\Omega(\omega, \omega^*, \mathbf{x}, y, z) = \alpha \cdot \Omega'(\omega, \omega^*, \mathbf{x}, y, z) + \beta \cdot \Omega''(\omega \backslash \omega^*) \tag{4}$$

where $\alpha, \beta$ are two non-negative tuning parameters to balance two terms. On top of the *behavioral regularizer* $\Omega'(\omega, \omega^*, \mathbf{x}, y, z)$, DELTA includes a term $\Omega''(\omega \backslash \omega^*)$ regularizing a subset of parameters that are privately owned by the target network $w$ only but not exist in the source network $w^*$. Specifically, $\Omega''(\omega \backslash \omega^*)$ constrains the $L^2$-norm of the private parameters in $\omega$, so as to improve the consistency of inner layer parameters estimation. Note that, when using $w^*$ as the initialization of $\omega$ for optimization, DELTA indeed adopts starting point as reference (SPAR) strategy (Li et al., 2018) to accelerate the optimization and gains better generalizability.

## 3.2 BEHAVIORAL REGULARIZATION

To regularize the behavior of the networks, DELTA considers the distance between the outer layer outputs of the two networks. Figure 1 illustrates the concepts of proposed method. Specifically, the outer layer of the network consists of a large set of convolutional filters. Given an input $\mathbf{x}_i$ (for $\forall 1 \leq i \leq n$ in training set), each filter generates a feature map. Thus, DELTA characterizes the outer layer output of the network model $z$ based on input $\mathbf{x}_i$ and parameter $\omega$ using a set of feature maps, such as $\mathrm{FM}_j(z, \omega, \mathbf{x}_i)$ and $1 \leq j \leq N$ for the $N$ filters in networks. In this way, the behavioral regularizer is defined as:

$$\Omega'(\omega, \omega^*, \mathbf{x}_i, y_i, z) = \sum_{j=1}^{N} \left( \mathrm{W}_j(z, \omega^*, \mathbf{x}_i, y_i) \cdot \| \mathrm{FM}_j(z, \omega, \mathbf{x}_i) - \mathrm{FM}_j(z, \omega^*, \mathbf{x}_i) ) \|_2^2 \right) \tag{5}$$

where $W_j(z, \omega^*, \mathbf{x}_i, y_i)$ refers to the weight assigned to the $j^{th}$ filter and the $i^{th}$ image (for $\forall 1 \leq i \leq n$ and $\forall 1 \leq j \leq N$) and the behavioral difference between the two feature maps, i.e., $\mathrm{FM}_j(z, \omega, \mathbf{x}_i)$ and $\mathrm{FM}_j(z, \omega^*, \mathbf{x}_i)$, is measured using their Euclid distance (denoted as $\| \cdot \|_2$).

In following sections, we are going to present (1) the design and implementation of feature map extraction $\text{FM}_j(z, \omega, \mathbf{x})$ for $1 \leq j \leq N$, as well as (2) the the attention model that assigns the weight $\text{W}_j(z, \omega^*, \mathbf{x}_i, y_i)$ to each labeled image and filter.

### 3.3 Feature Map Extraction from Convolution Layers

Given each filter of the network with parameter $\omega$ and the input $x_i$ drawn from the target dataset, DELTA first uses such filter to get the corresponding output based on $x$, then adopts Rectified Linear Units (ReLU) to rectify the output as a matrix. Further, DELTA formats the output matrices into vectors through concatenation. In this way, DELTA obtains $\text{FM}_j(z, \omega, \mathbf{x}_i)$ for $1 \leq j \leq N$ and $1 \leq i \leq n$ that have been used in Eq 5.

### 3.4 Weighting Feature Maps with Supervised Attention Models

In DELTA, the proposed regularizer measures the distance between the feature maps generated by the two networks, then aggregates the distances using non-negative weights. Our aim is to pay more attention to those features with greater capacity of discrimination through supervised learning. To obtain such weights for feature maps, we propose a supervised attention method derived from the backward variable selection, where the weights of features are characterized by the potential performance loss when removing these features from the network.

For clear description, following common conventions, we first define a convolution filter as follow. The parameter of a conv2d layer is a four-dimensional tensor with the shape of $(c_{i+1}, c_i, k_h, k_w)$, where $c_i$ and $c_{i+1}$ represent for the number of channels of the $i_{th}$ and $(i+1)_{th}$ layer respectively. $c_{i+1}$ filters are contained in such a convolutional layer, each of which with the kernel size of $c_i * k_h * k_w$, taking the feature maps with the size of $c_i * h_i * w_i$ of the i-th layer as input, and outputing the feature map with the size of $h_{i+1} * w_{i+1}$.

In particular, we evaluate the weight of a filter as the performance reduction when the filter is disabled in the network. Intuitively, removing a filter with greater capacity of discrimination usually causes higher performance loss. In this way, such channels should be constrained more strictly since a useful representation for the target task is already learned by the source task. Given the pre-trained parameter $\omega^*$ and an input image $x_i$, DELTA sets the weight of the $j^{th}$ channel, using the gap between the empirical losses of the networks on the labeled sample $(x_i, y_i)$ with and without the $j^{th}$ channel, as follow,

$$\text{W}_j(z, \omega^*, \mathbf{x}_i, y_i) = \text{softmax} \left( L(z(\mathbf{x}_i, \omega^{*\backslash \mathbf{j}}), y_i) - L(z(\mathbf{x}_i, \omega^*), y_i) \right) \tag{6}$$

where $\omega^{*\backslash \mathbf{j}}$ refers to the modification of original parameter $\omega^*$ with all elements of the $j^{th}$ filter set to zero (i.e., removing the $j^{th}$ filter from the network). We use softmax to normalize the result to ensure all weights are non-negative. The aforementioned supervised attention mechanism yields a filter a higher weight for a specific image if and only if the corresponding feature map in the pre-trained source network is with higher discrimination power — i.e., paying more attention to such filter on such image might bring higher performance gain.

Note that, to calculate $L(z(\mathbf{x}_i, \omega^{*\backslash \mathbf{j}}), y_i)$ and $L(z(\mathbf{x}_i, \omega^*), y_i)$ for supervised attention mechanism, we introduce a baseline algorithm $L^2\text{-}FE$ that fixes the feature extractor (with all parameters copied from source networks) and only trains the discriminators using the target task. The $L^2\text{-}FE$ model can be viewed as an adaption of the source network (weights) to the target tasks, without further modifications to the outer layer parameters. In our work, we use $L^2\text{-}FE$ to evaluate $L(z(\mathbf{x}_i, \omega^{*\backslash \mathbf{j}}), y_i)$ and $L(z(\mathbf{x}_i, \omega^*), y_i)$ using the target datasets.

## 4 Experiments and Results

We have conducted a comprehensive experimental study of the proposed DELTA method. Below we first briefly review the used datasets, followed by a description of experimental procedure and finally our observations.

## 4.1 DATASETS

We evaluate the performance of three benchmarks with different tasks: Caltech 256 for general object recognition, Stanford Dogs 120 for fine-grained object recognition, and MIT Indoors 67 for scene classification. For the first two benchmarks, we used ImageNet as the source domain and Places 365 for the last one.

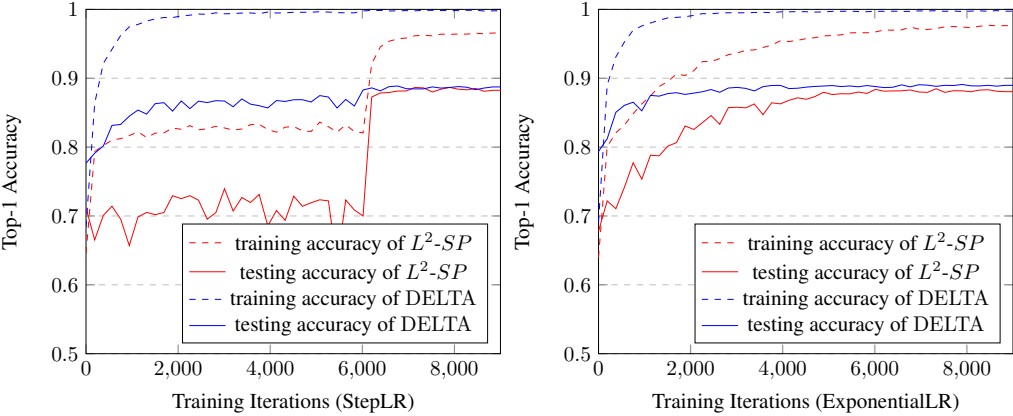

Figure 2: Learning curves of the proposed feature map based regularization(DELTA) compared with weight based regularization($L^2$-$SP$) on the Stanford Dog 120 benchmark using different methods to adjust the learning rate. StepLR: setting the learning rate to the initial value decayed by 0.1 after 6000 iterations (32 epochs for the Stanford Dogs dataset). ExponentialLR: setting the learning rate to the initial value decayed by 0.93 every epoch.

**Caltech 256.** Caltech 256 is a dataset with 256 object categories containing a total of 30607 images. Different numbers of training examples are used by researchers to validate the generalization of proposed algorithms. In this paper, we create two configurations for Caltech 256, which have 30 and 60 random sampled training examples respectively for each category, following the procedure used in (Li et al., 2018).

**Stanford Dogs 120.** The Stanford Dogs dataset contains images of 120 breeds of dogs from around the world. There are exactly 100 examples per category in the training set. It is used for the task of fine-grained image categorization. We do not use the bounding box annotations.

**MIT Indoors 67.** MIT Indoors 67 is a scene classification task containing 67 indoor scene categories, each of which consists of 80 images for training and 20 for testing. Indoor scene recognition is challenging because both spatial properties and object characters are expected to be extracted.

**Caltech-UCSD Birds-200-2011.** CUB-200-2011 contains 11,788 images of 200 bird species. Each species is associated with a wikipedia article and organized by scientific classification. Each image is annotated with bounding box, part location, and attribute labels. We use only classification labels during training. While part location annotations are used in a quantitative evaluation of show cases, to explain the transferring effect of our algorithm.

**Food-101.** Food-101 a large scale data set of 101 food categories, with 101,000 images, for the task of fine-grained image categorization. 750 training images and 250 test images are provided for each class. This dataset is challenging because the training images contain some amount of noise.

## 4.2 EXPERIMENTAL PROCEDURE

We implemented our method with ResNet-101 and Inception-V3 as the base networks. For experiment set up we followed almost the same procedure in (Li et al., 2018) due to the close relationship between our work and theirs. After training with the source dataset and before fine-tuning the network with the target dataset, we replace the last layer of the base network with random initialization in suit for the target dataset.

Table 1: Comparison of top-1 accuracy with different methods. $L^2\text{-}FE$: Using the pre-trained model as a feature extractor. Baselines: $L^2\text{-}FE$, $L^2$ and $L^2\text{-}SP$.

| ResNet-101 | $L^2\text{-}FE$ | $L^2$ | $L^2\text{-}SP$ | DELTA(w/o ATT) | DELTA |
|---|---|---|---|---|---|
| MIT Indoors 67 | $80.4 \pm 0.2$ | $83.7 \pm 0.3$ | $85.1 \pm 0.1$ | $85.3 \pm 0.2$ | $\mathbf{85.5 \pm 0.3}$ |
| Stanford Dogs 120 | $84.7 \pm 0.1$ | $83.3 \pm 0.2$ | $88.3 \pm 0.2$ | $88.3 \pm 0.2$ | $\mathbf{88.7 \pm 0.1}$ |
| Caltech 256-30 | $82.9 \pm 0.2$ | $84.7 \pm 0.3$ | $85.4 \pm 0.2$ | $85.7 \pm 0.3$ | $\mathbf{86.6 \pm 0.1}$ |
| Caltech 256-60 | $85.3 \pm 0.2$ | $87.2 \pm 0.3$ | $87.2 \pm 0.1$ | $87.6 \pm 0.2$ | $\mathbf{88.7 \pm 0.1}$ |
| CUB-200-2011 | $61.5 \pm 0.1$ | $78.4 \pm 0.1$ | $79.5 \pm 0.1$ | $78.9 \pm 0.1$ | $\mathbf{80.5 \pm 0.1}$ |
| Food-101 | $64.3 \pm 0.1$ | $85.3 \pm 0.1$ | $\mathbf{86.4 \pm 0.1}$ | $85.9 \pm 0.1$ | $86.3 \pm 0.2$ |
| Inception-V3 | $L^2\text{-}FE$ | $L^2$ | $L^2\text{-}SP$ | DELTA(w/o ATT) | DELTA |
| MIT Indoors 67 | $74.9 \pm 0.2$ | $74.8 \pm 0.4$ | $74.6 \pm 0.4$ | $76.9 \pm 0.3$ | $\mathbf{78.1 \pm 0.4}$ |
| Stanford Dogs 120 | $84.1 \pm 0.1$ | $88.6 \pm 0.2$ | $\mathbf{89.4 \pm 0.1}$ | $88.7 \pm 0.1$ | $88.7 \pm 0.1$ |
| Caltech 256-30 | $82.5 \pm 0.2$ | $83.6 \pm 0.3$ | $83.3 \pm 0.2$ | $83.4 \pm 0.3$ | $\mathbf{84.9 \pm 0.2}$ |
| Caltech 256-60 | $84.1 \pm 0.1$ | $85.8 \pm 0.3$ | $85.3 \pm 0.1$ | $85.1 \pm 0.2$ | $\mathbf{86.8 \pm 0.1}$ |
| CUB-200-2011 | $57.6 \pm 0.1$ | $74.3 \pm 0.2$ | $75.2 \pm 0.1$ | $74.5 \pm 0.1$ | $\mathbf{76.5 \pm 0.1}$ |
| Food-101 | $55.9 \pm 0.1$ | $76.9 \pm 0.2$ | $75.9 \pm 0.2$ | $76.2 \pm 0.2$ | $\mathbf{80.8 \pm 0.2}$ |

For ResNet-101, the input images are resized to 256*256 and normalized to zero mean for each channel, following with data augmentation operations of random mirror and random crop to 224*224. For Inception-V3, images are resized to 320*320 and finally cropped to 229*229. We use a batch size of 64. SGD with the momentum of 0.9 is used for optimizing all models. The learning rate for the base model starts with 0.01 for ResNet-101 and 0.001 for Inception-V3, and is divided by 10 after 6000 iterations. The training is finished at 9000 iterations. We use five-fold cross validation for searching the best configurations of the hyperparameter $\alpha$ for each experiment. The hyperparameter $\beta$ is fixed to 0.01. As was mentioned, our experiments compared DELTA to several key baseline algorithms including $L^2$, $L^2\text{-}SP$ (Li et al., 2018), and $L^2\text{-}FE$ (see also in Section 3.4), all under the same settings. Each experiment is repeated five times. The average top-1 classification accuracy and standard division is reported.

## 4.3 RESULTS AND COMPARISONS

In Fig 2 we plotted a sample learning curve of training with different regularization techniques. Comparing these regularization techniques, we observe that our proposed DELTA shows faster convergence than the simple $L^2\text{-}SP$ regularization with both step decay (StepLR) and exponential decay (ExponentialLR) learning rate scheduler. In addition, we find that the learning curve of DELTA is smoother than $L^2\text{-}SP$ and it is not sensitive to the learning rate decay happened at the 6000th iteration when using StepLR.

In Table 1 we show the results of our proposed method DELTA with and without attention, compared to the baseline of $L^2\text{-}SP$ reported in (Li et al., 2018) and also the naive $L^2\text{-}FE$ and $L^2$ methods. We find that on some datasets, fine-tuning using $L^2$ normalization does not perform significantly better than directly using the pre-trained model as a feature extractor($L^2\text{-}FE$), while $L^2\text{-}SP$ outperforms the naive methods without SPAR. We observe that greater benefits are gained using our proposed attention mechanism.

Data augmentation is a widely used technique to improve image classification. Following (Li et al., 2018), we used a simple data augmentation method and a post-processing technique. First, we keep the original aspect ratio of input images by resizing them with the shorter edge being 256, instead of ignoring the aspect ratio and directly resizing them to 256*256. Second, we apply 10-crop testing to further improve the performance. In Table 2, we documented the experimental results using these technique with different regularization methods. We observe a clear pattern that with additional data augmentation, all the three evaluated methods $L^2$, $L^2\text{-}SP$, DELTA have improved classification accuracy while our method still delivers the best one.

Table 2: Comparing top-1 accuracy using data augmentation for three regularization methods.

| ResNet-101 | $L^2$ | $L^2$-$SP$ | DELTA |
|---|---|---|---|
| MIT Indoors 67 | $84.4 \pm 0.5$ | $85.2 \pm 0.3$ | $\mathbf{85.9 \pm 0.3}$ |
| Stanford Dogs 120 | $85.7 \pm 0.2$ | $90.8 \pm 0.2$ | $\mathbf{91.2 \pm 0.2}$ |
| Caltech 256-30 | $85.1 \pm 0.4$ | $86.4 \pm 0.2$ | $\mathbf{87.1 \pm 0.2}$ |
| Caltech 256-60 | $87.4 \pm 0.2$ | $88.3 \pm 0.1$ | $\mathbf{89.1 \pm 0.1}$ |
| CUB-200-2011 | $81.7 \pm 0.2$ | $82.3 \pm 0.2$ | $\mathbf{82.6 \pm 0.2}$ |
| Food-101 | $86.7 \pm 0.1$ | $87.2 \pm 0.2$ | $\mathbf{87.5 \pm 0.1}$ |
| Inception-V3 | $L^2$ | $L^2$-$SP$ | DELTA |
| MIT Indoors 67 | $75.5 \pm 0.4$ | $76.5 \pm 0.3$ | $\mathbf{78.7 \pm 0.3}$ |
| Stanford Dogs 120 | $91.2 \pm 0.1$ | $91.9 \pm 0.1$ | $\mathbf{92.1 \pm 0.1}$ |
| Caltech 256-30 | $84.7 \pm 0.2$ | $84.5 \pm 0.2$ | $\mathbf{85.5 \pm 0.2}$ |
| Caltech 256-60 | $86.1 \pm 0.2$ | $86.0 \pm 0.1$ | $\mathbf{87.0 \pm 0.2}$ |
| CUB-200-2011 | $76.3 \pm 0.3$ | $76.3 \pm 0.2$ | $\mathbf{77.6 \pm 0.3}$ |
| Food-101 | $78.2 \pm 0.1$ | $77.2 \pm 0.2$ | $\mathbf{82.1 \pm 0.2}$ |

## 4.4 A CASE STUDY AND DISCUSSIONS

To better understand the performance gain of DELTA we performed an experiment where we analyzed how parameters of the convolution filters change after fine-tuning. Towards that purpose we randomly sampled images from the testing set of Stanford Dogs 120. For ResNet-101, which we use exclusively in this paper, we grouped filters into stages as described in (he et al., 2016). These stages are conv2_x, conv3_x, conv4_x, conv5_x. Each stage contains a few stacked blocks and a block is a basic inception unit having 3 conv2d layers. One conv2d layer consists of a number of output filters. We flatten each filter into a one dimension parameter vector for convenience. The Euclidian distance between the parameter vectors before and after fine-tuning is calculated. All distances are sorted as shown in Figure 3.

We observed a sharp difference between the two distance distributions. Our hypothesis of possible cause of the difference is that simply using $L^2$-$SP$ regularization all convolution filters are forced to be similar to the original ones. Using attention, we allow "unactivated" convolution filters to be re-used for better image classification. About 90% parameter vectors of DELTA have larger distance than $L^2$-$SP$. We also observe that a small number of filters is driven very far away from the initial value (as shown at the left end of the curves in Figure 3). We call such an effect as "unactivated channel re-usage".

To further understand the effect of attention and the implication of "unactivated channel re-usage", we "attributed" the attention to the original image to identify the set of pixels having high contributions in the activated feature maps. We select some convolution filters on which the source model (the initialization before fine-tuning) has low activation. For the convenience of analyzing the effect of regularization methods, each element $a_i$ of the original activation map is normalized with

$$a_i = (a_i - \min_j a_j)/(\max_j a_j - \min_j a_j),$$

where the $\min$ and $\max$ terms in the formula represent for the minimum and maximum value of the whole activation map respectively. Activation maps of these convolution filter for various regularization method are presented on each row.

As shown in Figure 4, our first observation is that without attention, the activation maps from DELTA in different images are more or less the same activation maps from other regularization methods. This partially explains the fact that we do not observe significant improvement of DELTA without attention.

Using attention, however, changes the activation map significantly. Regularization of DELTA with attention show obviously improved concentration. With attention (the right-most column in Figure 4), we observed a large set of pixels that have high activation at important regions around the head of the animals. We believe this phenomenon provides additional evidence to support our intuition of "unactivated channel re-usage" as discussed in previous paragraphs.

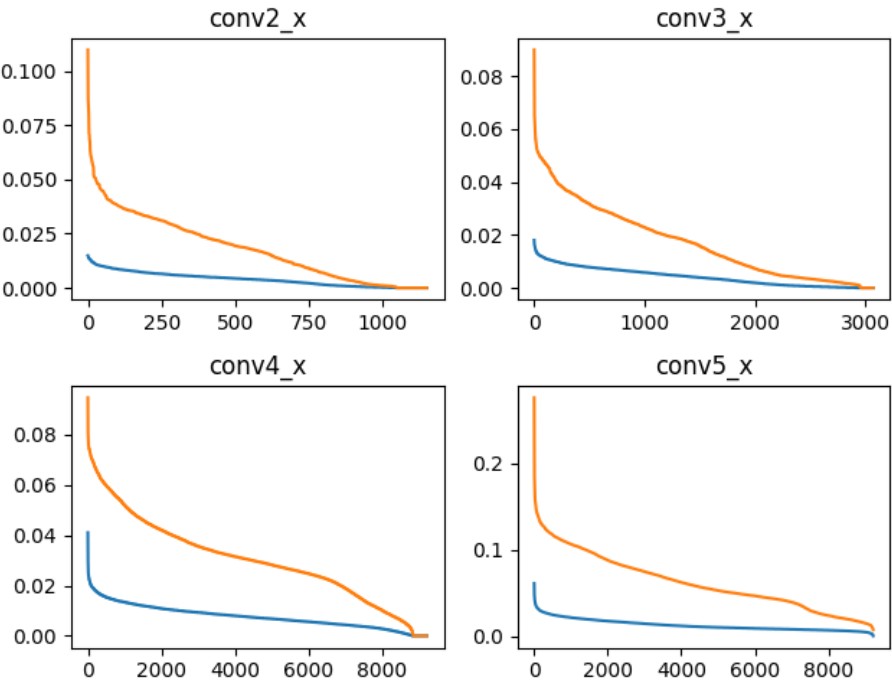

Figure 3: Distribution of the distance of parameters from the starting point. In ResNet-101, conv2_x, conv3_x, conv4_x, conv5_x represent for four main stages each of which has stacked convolution layers. The blue line represents for the result of $L^2$-$SP$, and the orange line for DELTA.

Table 3: Comparing average activations on 15 discriminate parts of CUB-200-2011 datasets for different regularization methods.

|  | SRC | $L^2$ | $L^2$-$SP$ | DELTA(w/o ATT) | DELTA |
|---|---|---|---|---|---|
| Average Activations | 5.298 | 5.392 | 6.258 | 6.241 | **6.367** |

In addition, we included new statistical results of activations on part locations of CUB-200-2011 supporting the above qualitative cases. The CUB-200-2011 datasets defined 15 discriminative parts of birds, e.g. the forehead, tail, beak and so on. Each part is annotated with a pixel location representing for its center position if it is visible. So for each image, we got several key points which are very important to discriminate its category. Using all testing examples of CUB-200-2011, we calculated normalized activations on these key points of these different regularization methods. As shown in Table 3, DELTA got the highest average activations on those key points, demonstrating that DELTA focused on more discriminate features for bird recognition.

## 5 CONCLUSION

In this paper, we studied a regularization technique that transfers the behaviors and semantics of the source network to the target one through constraining the difference between the feature maps generated by the convolution layers of source/target networks with attentions. Specifically, we designed a regularized learning algorithm DELTA that models the difference of feature maps with attentions between networks, where the attention models are obtained through supervised learning. Moreover, we further accelerate the optimization for regularization using start point as reference (SPAR). Our extensive experiments evaluated DELTA using several real-world datasets based on commonly used convolutional neural networks. The experiment results show that DELTA is able to significantly outperform the state-of-the-art transfer learning methods.

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

# A    APPENDIX

Examples with different regularization methods from Stanford Dogs and CUB-200-2011.

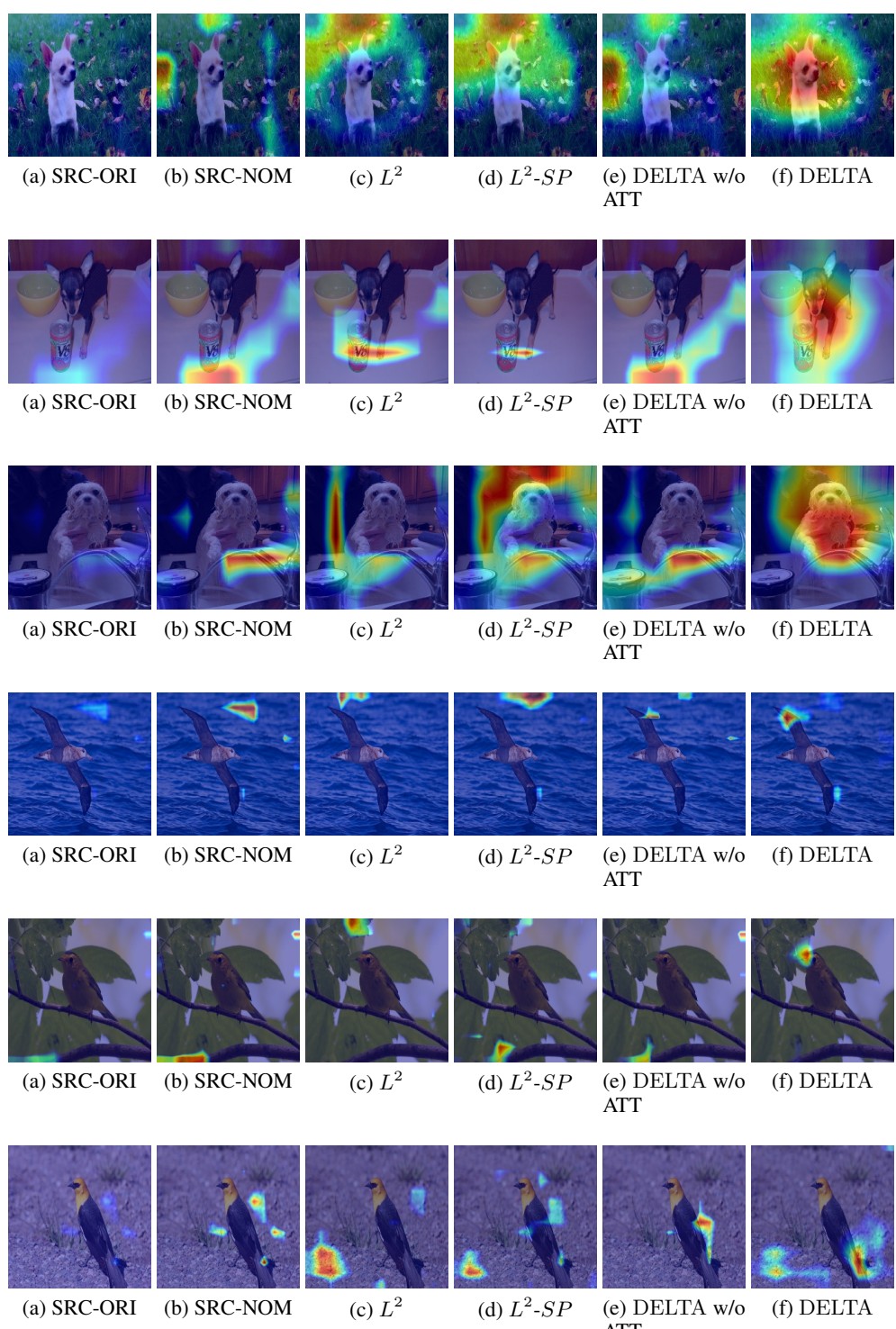

Figure 4: Illustration of the effect of the attention mechanism for fine-tuning. DELTA w/o ATT: DELTA without Attentions.

