# OpenReview forum: "DELTA: DEEP LEARNING TRANSFER USING FEATURE MAP WITH ATTENTION FOR CONVOLUTIONAL NETWORKS"
_ICLR.cc/2019/Conference_

### Official Review · AnonReviewer3 · 2018-10-29
**The paper proposes a normalization procedure improving transfer learning by considering together the output layers of the source and target deep nets. The contribution is rather limited, the experiments are reasonable but not always complete, the use of language satisfactory but can be improved.**

**Rating:** 6
**Confidence:** 4

**Review:**

This is a reasonable paper based on a simple intuition. The authors have noticed that some of the state of the art methods (they use Li et al - ICML18 as the main reference) are using only some simple normalization for improving the transfer learning and as such they propose preserving the outer layer output of the target network and aligning it with the one of the source network. On top of that they also propose modeling the difference of feature maps considering an attention mechanism obtain through supervised learning.

The idea in itself is interesting and valuable. However, I have had some difficulty in understanding precisely how the "behavior" is really regularized. While I understand what is depicted in Figure 1 I'm not completely sure this really means that the network behavior is regularized rather than simply correlating the two outputs. In the evaluation, the authors present in Figure 4 some qualitative examples but I would have expected to see some quantitative evaluation of this. I would have liked to see experiments on some larger datasets that are commonly used in computer vision (e.g., Caltech 256 is rather old even if it has been used in Li et al.). The quantitative results in Table 1 and 2 indicate some slight improvement but I'm not completely convinced that this is really significant in the end. The results in Figure 4 tend to show that with the attention mechanism there is a central bias and most of the results tend to be concentrated on the center of the image (in this case the result might also be correct but the examples presented are not too eloquent).

---

> ### Author Response · Authors · 2018-11-10
> **Thanks for the review**
>
> Thank you for your review and encouraging comments. We summarize and respond Reviewer 3’s concerns as follow.
>
> Q1. I have had some difficulty in understanding precisely how the "behavior" is really regularized. While I understand what is depicted in Figure 1 I'm not completely sure this really means that the network behavior is regularized rather than simply correlating the two outputs
>
> Response: Thanks for your comments. Yes, one intuition of our regularizer for deep transfer learning is to ensure the “behavior” of outer layers of the target network being similar to the source one. In this context,  “behavior” suggests functional preservation where similar inputs will produce similar outputs.  To measure similarity, however, is difficult.  Using feature map is a reasonable choice but such choice has many problems. First of all, feature maps/outer layer outputs are large with noise and redundancy. We have to understand which parts of feature maps (outputs) would help for the classification on the target task (rather than source task). In this way, an attention mechanism has been proposed to re-weight the output of each filter in out layers. Then, with the obtained attention, we characterize the divergence of feature maps/outputs between source/target networks as the attention-weighted squared-Euclidian distance between the feature maps. In this way, the regularizer to “correlate” the output has been designed. We further leverage some optimization paradigms, such as starting point as reference, to accelerate the deep transfer learning with better efficiency and effectiveness. Please refer to our methodologies section for details.
>
>
> Q2. the authors present in Figure 4 some qualitative examples but I would have expected to see some quantitative evaluation of this. I would have liked to see experiments on some larger datasets that are commonly used in computer vision
>
>
> Response: Thanks for your comments. Yes, eventually, handling large target dataset is an interesting topic in deep transfer learning research. Actually, we usually assume the target dataset size is relevantly small. As was stated in the first paragraph of the first section, the deep transfer learning studied here is originally motivated by the need of deep learning from small datasets. In such cases, CNNs are not be able to learn generalizable features from the small training sets, while weights of pre-trained networks with rich features learned from large datasets are assumed to help. To address your comments, we will include the results of supplementary experiments on both new architecture (inception v3) and datasets (CUB-200-2011[1], Food-101[2]). The new results will be reported in our incoming revision.
>
> [1] C.Wah,S.Branson,P.Welinder,P.Perona,andS.Belongie. The caltech-ucsd birds-200-2011 dataset. California Institute of Technology, 2011. 6, 7, 8
> [2] L. Bossard, M. Guillaumin, and L. Van Gool. Food-101–mining discriminative components with random forests. In
> ECCV, 2014. 6, 8
>
>
> Q3. The results in Figure 4 tend to show that with the attention mechanism there is a central bias and most of the results tend to be concentrated on the center of the image
>
> Response: Thanks for your comments for catching the possible location bias. In our experiments we do not find evidence that our attention mechanism has a strong central bias. It pays attention to the parts of feature maps that are with discriminant capacities. The attention illustrated on dog images is just a coincidence. In our revised version, we plan to present more attention results to avoid the impression of central bias.

---

### Official Review · AnonReviewer2 · 2018-11-01
**Interesting approach but not yet clearly demonstrating a significant boost in performance**

**Rating:** 6
**Confidence:** 4

**Review:**

Authors present a new regularisation approach named DELTA (Deep Learning Transfer using feature map with attention). What it does is preserving the outer layer outputs of the target network (in a transfer learning scenario) instead of constraining the weights of the neural network. I am not sure how this approach helps preserve the semantics. Authors state that the distance between source/target networks is characterised by DELTA using their outer layer outputs. This distance is then used in the loss function and through back-propagation incorporates knowledge from the source network. The results demonstrate some marginal improvement in the datasets used when compared with L^2 and L^2-SP.
More importantly I think the paper needs some attention in its format as the concepts are not very clear. It has some elements of novelty but not yet there.

Authors have addressed most of my issues and hence I have revised my decision.

---

> ### Author Response · Authors · 2018-11-10
> **Thanks for the review**
>
> Thank you for your review and constructive comments. We summarize the major concerns of Reviewer 2 as following:
>
> Q1. There needs evidence of (statistically) significant performance boosting
>
> Response: Thanks for your comments. In our experiments, we run DELTA and other baselines for multiple times under various settings, then estimate the mean accuracy with error bars in Tables 1. We further compare the worst-case performance (the lowest accuracy) of DELTA to the best-case performance (the highest accuracy) of rest baselines under each setting. The results suggest that DELTA can always perform better than the baseline algorithms, as its worst-case performance is marginally better than the best-case performance of baselines in our experiments. To address your comments, we will include supplementary experiments on some new neural architectures (inception v3) and datasets (CUB-200-2011[1], Food-101[2]). The new results will be compared and reported in our incoming revision.
>
> Q2. What is the major contributions made in this paper.
>
> Response:  Thanks for your comment. Our key innovation is the concept that we call "unactivated channel re-usage”. Specifically we demonstrate by experiments and case studies that, when we perform transfer learning in CNN, some of the convolution channels are useless in transfer learning. In contrast to the common belief that lower level convolution channels are for common feature extraction, higher level channels are for task-specific feature extraction, we demonstrate that there are channels that are useful (and useless) at all different levels.
>
> Our technical contribution is to find an approach to identify those “transferable channels” and preserve them through regularization and identify those “untransferable channels” and reuse them, using an attention mechanism with feature map regularization.
>
> Compared to existing deep transfer learning paradigms reusing the weights of outer layers of source networks, DELTA intends to constrain on the difference of the outputs (e.g., feature maps) rather than the weights between source/target networks. Compared to the knowledge distillation alike solution (if they are adopted for transfer learning), DELTA proposes a novel attention mechanism to make target network stay focused on the important features with high discriminant powers for knowledge transfer. To the best of knowledge, we are the first to regularize the divergence of the feature maps outputted by the outer layers of the source/target networks, with attention mechanisms, for deep transfer learning.
>
> Using image classification and case studies we demonstrate the usefulness of our insights. In addition, we plan to further demonstrate the utility of the proposed methods by adding supplementary experiments on both new architecture(inception v3) and datasets(CUB-200-2011[1], Food-101[2]). Those results will be reported in our new version.
>
> [1] C.Wah,S.Branson,P.Welinder,P.Perona,andS.Belongie. The caltech-ucsd birds-200-2011 dataset. California Institute of Technology, 2011. 6, 7, 8
> [2] L. Bossard, M. Guillaumin, and L. Van Gool. Food-101–mining discriminative components with random forests. In ECCV, 2014. 6, 8

---

### Official Review · AnonReviewer1 · 2018-11-03
**main contributions: bringing in the concept from teacher-student, same task to different task in transfer learning by modifying the weights of outer layers**

**Rating:** 7
**Confidence:** 3

**Review:**

Summary
The paper describes using the technique of modifying the weights for the outer layers, used in teacher-student network for same task, to transfer learning for different tasks by modifying the loss function and pre-training using target network labels to emphasize the neurons that are considered important for prediction. The technique seems to be no more/slightly better than the Lsquare SP, but exceeds when used with attention.

Improvements
- the amount of training time needed to pre-train using the L-square FE and target labels should be mentioned as it seems that for large network, and large data, this can be a factor
- The choice of Resnet, at least one of the more recent networks for object detection (Inception, YOLO etc.)  would be a good add

---

> ### Author Response · Authors · 2018-11-10
> **Thanks for the review**
>
> Thank you for your review and encouraging comments. We summarize Reviewer 1’s major concerns as following two questions and we try to respond these two answers accordingly.
>
> Q1. “the amount of training time needed to pre-train using the L-2 FE and target labels should be mentioned as it seems that for large network, and large data, this can be a factor”
>
> Response: Thanks for the comment. We totally agree that the time-consumption of L2-FE adaption and attention learning should be considered as the overhead of our method. Indeed, the time spent by L2-FE adaption and attention learning is no more than 50% of overall training time.  For example, DELTA (w/o Attention) requires 139 minutes on Caltech30 task transferring from Resnet-101 pre-trained model, while DELTA (with Attention) consumes 197 minutes (42% more than DELTA w/o Attention which doesn’t need L2-FE adaption and attention learning). Furthermore, L2-SP takes124 minutes and L2 spends 115 minutes on the same task in the same settings. It is thus reasonable to conclude the extra time consumption on L2-FE adaption and attention learning does not add a significant overhead to common deep transfer learning practices.
>
> When we further breakdown such time overhead, we found that the major overhead is due to the attention learning, where for each filter forward inference is needed to estimate the contribution of such filter to the overall accuracy. It should not be a significant performance bottleneck even for a large dataset, as the number of filters needed to evaluate might be fixed with given scratch for transfer learning.  Note that one key contribution made in this manuscript is to improve the deep transfer learning via feature-map based regularization through introducing attention mechanism. All in all, many thanks for your comments. We are revising the manuscript, including the discussion on time consumption, accordingly.
>
> Q.2 The choice of Resnet, at least one of the more recent networks for object detection (Inception, YOLO etc.)  would be a good add
>
> Response: Thanks for comments. We agree to incorporate more results and neural network architectures. We are revising the manuscript with supplementary experiments on both new architecture (inception v3) and datasets(CUB-200-2011[1], Food-101[2]). The results of above experiments will be reported in our new version.
>
> [1] C.Wah,S.Branson,P.Welinder,P.Perona,andS.Belongie. The caltech-ucsd birds-200-2011 dataset. California Institute of Technology, 2011. 6, 7, 8
> [2] L. Bossard, M. Guillaumin, and L. Van Gool. Food-101–mining discriminative components with random forests. In ECCV, 2014. 6, 8

---

> > ### Comment · AnonReviewer1 · 2018-11-13
> > **Comparison to similar literature published previously**
> >
> > Can you also explain similarity of this method with "PAYING MORE ATTENTION TO ATTENTION: IMPROVING THE PERFORMANCE OF CONVOLUTIONAL NEURAL NETWORKS VIA ATTENTION TRANSFER" published in ICLR 17 in particular the activation based attention transfer in the context of teacher-student network, and explain how the main idea in your submission is step above that?

---

> > > ### Author Response · Authors · 2018-11-14
> > > **Reply to "Comparison to similar literature published previously"**
> > >
> > > Thank you very much for the comment. The work  (Zagoruyko & Komodakis, 2016) was cited in our original submission. We fully agree that it is highly  relevant to our paper. The connections between these works have been briefly stated at the last paragraph of related work in our original paper. Below we would like to elaborate the major differences.
> > >
> > > We have different goals. Zagoruyko & Komodakis aim to obtain a different (and smaller) network to perform the same task.  While DELTA use a similar attention mechanism, we aim to obtain a network to perform a different task. With the different goal, we have to adopt a similar but distinct approach for attention to enable efficient knowledge transfer.
> > >
> > > Methodologies. Our definition of attention is based on the discriminant capacity of each filter in the network trained on the source domain as applied to the target domain.  In this approach, we are able to assign the weights to the regularizer for EACH filter for optimizing knowledge transfer. With such attention (accuracy contribution of each filter), DELTA assign higher weights to the useful filters and re-use filters that do not classify the target data well (i.e., to make those filter  freely optimized).
> > >
> > > On the other hand, Zagoruyko & Komodakis defined attention as the data representation obtained by teacher networks, where student networks are guided to obtain similar representation on the same data. Attention transfer considers attention as an objective that student needs to learn from teacher. It implements attention as the activation map, i.e, the aggregation of feature maps generated by all filters (in the same map) while we assign weight to each filter so that some of the filters can be re-used for new tasks.

---

> > ### Comment · AnonReviewer1 · 2018-11-16
> > **Re: Thanks for the review**
> >
> > Few follow-ups:
> >
> > 1. Is each filter in each layer done independently or we are looking at combinations (through the succeeding layers)?
> > 2. On this being not an issue for large data set, can you please put some numbers for Resnet 101 # of filters considered in eq. 6, times the number of instances in the train set used, for all the train datasets considered?

---

> > > ### Author Response · Authors · 2018-11-18
> > > **Reply to "Few follow-ups"**
> > >
> > > Thank you for your review.
> > >     1. Each filter (channel) is independently evaluated. Though it is time consuming, we only need to evaluate all these filters for one time, prior to the transfer learning.
> > >     2. We considered filters(we use output channels indeed because we constrained the behaviors) of layer conv2_x, conv3_x, conv4_x, conv5_x. They are with 256, 512, 1024, 2048 channels separately,  totally 3840 channels. We used 256 instances(four mini batches) in the training set for all the training datasets.

---

### Author Response · Authors · 2018-11-10
**We are running additional experiments as requested and will continue update our responses.**

We want to thank all the anonymous reviewers for your hard work and the insightful comments. As you may see below in our response, we are running additional experiments as requested. We decide to first provide our response based on the data that we have and we will continue update our responses. Thank you for your patience and we look forward to your further comments and discussions.

---

### Author Response · Authors · 2018-11-22
**We have updated our paper.**

Thank all the anonymous reviewers for your insightful comments and patience. We have updated our paper with the following modifications.
    1. We added two benchmark datasets, Caltech-UCSD Birds-200-2011 and Food-101,  to provide additional evaluation of transfer learning algorithms.  They were introduced in Section 4.1.
    2. We added supplementary experiments with a new architecture Inception-V3 and also the new datasets. We updated training details in Section 4.2. Results at Table 1 and 2 were updated with the additional experiments.
    3. We presented additional case studies from the CUB-200-2011 datasets. We also did a quantitative evaluation using part location annotations of the CUB-200-2011 datasets, showing that DELTA captured discriminative features for bird recognition. That provides further evidences for the effect of our key insight on "unactivated channel re-usage".

---

### Meta-Review · Area_Chair1 · 2018-12-11
**An interesting approach to transfer learning by focusing on relevant channels**

**Confidence:** 4
**Recommendation:** Accept (Poster)

**Metareview:**

This paper argues that each layer of a network may have some channels useful for and some not useful for transfer learning. The main contribution is an approach which identifies the useful channels through an attention based mechanism. The reviewers agree that this work offers a valuable new approach that offers modest improvements over prior work.

The authors should take care to refine their definition of behavior regularization, including/expanding on the discussion from the rebuttal phase. The authors are also encouraged to experiment with other architecture backbones and report both overall performance as well as run time for learning with the larger models.